# Involvement of Inflammatory Cytokines, Renal NaPi-IIa Cotransporter, and TRAIL Induced-Apoptosis in Experimental Malaria-Associated Acute Kidney Injury

**DOI:** 10.3390/pathogens13050376

**Published:** 2024-05-01

**Authors:** Gustavo Martins Simião, Kleber Simônio Parreira, Sandra Gabriela Klein, Flávia Batista Ferreira, Fernanda de Souza Freitas, Eduardo Ferreira da Silva, Neide Maria Silva, Murilo Vieira da Silva, Wânia Rezende Lima

**Affiliations:** 1Faculty of Health Sciences, Federal University of Rondonopolis, Rondonópolis 78736-900, MT, Brazil; gustavo.simiao@aluno.ufr.edu.br; 2Institute of Biotechnology, Federal University of Catalao, Catalão 75706-881, GO, Brazil; fernandade@discente.ufcat.edu.br (F.d.S.F.); eduardo_silva@discente.ufg.br (E.F.d.S.); 3Laboratory of Biotechnology in Experimental Models, Federal University of Uberlandia, Uberlândia 38410-337, MG, Brazil; klein.sandra@ufu.br (S.G.K.); flaviabatistaf@ufu.br (F.B.F.); murilo.vieira@ufu.br (M.V.d.S.); 4Institute of Biomedical Sciences, Federal University of Uberlandia, Uberlândia 38405-318, MG, Brazil; nmsilva@ufu.br

**Keywords:** malaria, cytokine, apoptosis, kidney

## Abstract

The murine model of experimental cerebral malaria (ECM) induced by *Plasmodium berghei* ANKA was used to investigate the relationship among pro-inflammatory cytokines, alterations in renal function biomarkers, and the induction of the TRAIL apoptosis pathway during malaria-associated acute kidney injury (AKI). Renal function was evaluated through the measurement of plasma creatinine and blood urea nitrogen (BUN). The mRNA expression of several cytokines and NaPi-IIa was quantified. Kidney sections were examined and cytokine levels were assessed using cytometric bead array (CBA) assays. The presence of glomerular IgG deposits and apoptosis-related proteins were investigated using in situ immunofluorescence assays and quantitative real-time PCR, respectively. NaPi-IIa downregulation in the kidneys provided novel insights into the pathogenesis of hypophosphatemia during CM. Histopathological analysis revealed characteristic features of severe malaria-associated nephritis, including glomerular collapse and tubular alterations. Pro-inflammatory cytokines, such as TNF-α, IL-1β, and IL-6, were upregulated. The TRAIL apoptosis pathway was significantly activated, implicating its role in renal apoptosis. The observed alterations in renal biomarkers and the downregulation of NaPi-IIa shed light on potential mechanisms contributing to renal dysfunction in ECM. The intricate balance between pro- and anti-inflammatory cytokines, along with the activation of the TRAIL apoptosis pathway, highlights the complexity of malaria-associated AKI and provides new therapeutic targets.

## 1. Introduction

According to the World Malaria Report 2022, there were approximately 247 million new clinical cases of malaria due to *Plasmodium* and about 619,000 deaths world-wide in 2021 [1]. Malaria is endemic in 84 countries across Africa, Asia, and Central and South America. *Plasmodium vivax* is responsible for a large number of cases in both Central and South America, the Asia-Pacific (especially Southeastern Asia) region and the Eastern Mediterranean, whereas *P. falciparum* is largely distributed worldwide and causes severe malaria, leading to death [2,3]. Cerebral malaria (CM) is a severe complication developed by patients with *P. falciparum* malaria. CM is often accompanied by multi-organ complications, including central nervous system dysfunction, liver dysfunction, respiratory failure and acute kidney failure [4]. Indeed, renal failure sets in up to 40% and 60% of adults and children with CM, respectively [5,6]. Hypophosphatemia and phosphaturia can also be often identified in adults and children with severe malaria, although severe hypophosphatemia seems not to be associated with any deleterious effect in children [7].

The pathogenesis of CM has been extensively investigated in an experimental murine malaria model, known as experimental cerebral malaria (ECM), in which *Plasmodium berghei* ANKA-infected mice develop a set of clinical manifestations that resemble those of falciparum malaria. The *P. berghei* ANKA (PbA) infection of susceptible mouse strains, including C57BL/6 and CBA, causes a neurological syndrome characterized by paralysis, ataxia, convulsion and coma, with animals dying within two weeks after infection [8]. Some studies demonstrated a crucial role of parasitic red blood cell (iRBC) accumulation in the brain for the development of ECM [9,10,11,12]. These authors reported that iRBC sequesters in various organs, including the brain and kidney, and was associated with the onset of ECM. Nevertheless, a recent study demonstrated that experimental malaria acute kidney injury (AKI) occurs regardless of parasite sequestration, although the work was carried out in an uncomplicated murine malaria model [13]. The complete understanding of the pathogenic mechanisms of AKI in malaria is still lacking. Several pathological processes, including endothelial dysfunction and immune-mediated damage, as well as parasite sequestration, may impact the kidneys. A remarkable feature of malaria infection is intravascular hemolysis of iRBCs. This process results in the release of both host and parasite-derived molecules, which can elicit inflammatory responses [14].

Severe malaria patients have been found to produce high levels of systemic proinflammatory cytokines such as IFN-γ, IL-1β, IL-6, and TNF-α. Undoubtedly, the potent upregulation of these inflammatory cytokines in mice and humans with CM is a contributing factor not only to severe malaria pathogenesis but to malaria-associated AKI as well [15,16,17,18,19]. Certainly, macrophage infiltrates have been identified within the glomerulus of PbA-infected mice. This infiltration coincided with a significant increase in the expression of proinflammatory cytokines in kidney tissue, specifically TNF-α, IFN-γ and IL-6 [9,20]. Both TNF-α and IL-6 signaling play pivotal roles in amplifying inflammatory responses and triggering endothelial activation [21,22]. The TNF pathway, in particular, has demonstrated the ability to induce alterations in glomerular endothelial permeability. Endothelial activation results in the elevated expression of cell adhesion molecules on endothelial cell surfaces. This, in turn, could enhance iRBC sequestration and leukocyte infiltration into the kidneys, thereby contributing to the development of AKI.

Past studies have demonstrated immunoglobulin deposition in the glomeruli of mice infected with *Plasmodium* species and patients with severe malaria. This phenomenon is often associated with the sequestration of iRBCs in the microvasculature, including the glomerular capillaries [23,24,25,26]. Accordingly, sequestration can lead to the activation of the immune system and glomerular deposition of immunoglobulins, particularly immune complexes containing IgM and IgG. The deposition of immune complexes in glomeruli can trigger local inflammatory responses and contribute to the pathogenesis of glomerular and tubular injuries [27,28].

Consistent with AKI, pathological lesions such as acute interstitial nephritis and glomerulonephritis observed in patients and mice with malaria strongly suggest that *Plasmodium* infection leads to renal dysfunction [18,20,29,30,31]. Furthermore, histopathological changes in the kidney of patients with malaria-associated AKI have been attributed to events of tubular autophagy, necrosis and apoptosis [28,32].

The role of the TRAIL apoptosis pathway in some kidney diseases has been studied and TRAIL has been shown to be involved in glomerular and tubular injury. For instance, TRAIL activates caspase-3 (CASP3) and -8 (CASP8) and initiates apoptosis and/or proinflammatory effects in the renal parenchyma by binding to death receptors during ischemia reperfusion injury (IRI) and lupus nephropathy [33,34].

Although recent efforts have been made to elucidate the role of TRIAL apoptotic pathway in malaria-associated AKI, its cellular and molecular mechanisms still remain poorly understood. Therefore, the aim of the present study was to investigate the protein and mRNA expression levels of relevant inflammatory cytokines, the renal NaPi-IIa cotransporter and the TRAIL-induced apoptosis mediators TRIAL, CASP3 and CASP8 in both the blood and kidney of experimental mice affected by malaria-associated AKI. 

## 2. Materials and Methods

### 2.1. Animals

Male C57BL/6 mice aged 8 to 12 weeks and weighing 20–25 g were utilized in this study. The animals were provided by the Central Animal Facility of the Federal University of Uberlandia (UFU). They were kept in an environment with a controlled temperature (22 ± 2 °C) and housed in cages under 12 h day/night cycles with bedding changed frequently. Mice were given ad libitum access to food and water until the assays.

### 2.2. Mouse Infection and Analysis of Mortality, Morbidity, Parasitemia, and RMCBS Curves

Eight mice were infected with a single dose of 200 µL of saline solution containing 10^5^ erythrocytes infected with PbA via intraperitoneal (i.p.) injection. The control group was treated with a saline solution only (vehicle). For the analysis of mortality, morbidity, and rapid murine coma and behavior scale (RMCBS), the animals were monitored for up to 20 days after infection or treatment with the vehicle daily observations including blood smears, weight measurements, mortality assessments, and RMCBS scoring.

A drop of blood was collected from the animals via tail vein, and the parasitemia curve was determined by counting iRBCs. Thick blood smears were stained using Instant Prov Rapid hematological staining (NewProv, Pinhais, Brazil) and evaluated under a light microscope. The RMCBS is a quantitative scale that evaluates ten parameters (gait, balance, motor performance, body position, limb strength, touch escape, pinna reflex, toe pinch, aggression and grooming), and each parameter is scored 0 to 2, with a 0 score correlating with the lowest function and a 2 score the highest. An animal can achieve an accumulative score of 0 to 20 at each time point; that is, the lower the score, the higher the cognitive and motor dysfunction on the scale [35,36]. During the first 90 s of assessment, the mouse is placed in the top left corner of an observation box with a grid floor and is assessed for hygiene-related behavior, gait, body position, exploratory behavior, and balance. In the subsequent 90 s, the mouse is assessed for reflexes, limb strength, and self-preservation actions [36].

On day 7 after infection, the mice were intraperitoneally injected with an anesthetic solution containing ketamine (Syntec Brazil Ltda, Barueri, Brazil) and xylazine (Schering-Plough Coopers, São Paulo, Brazil). Serum samples were collected from the retro-orbital plexus, followed by euthanasia by cervical dislocation. Serum samples were used for cytometry bead array (CBA) analysis. Brain and kidney tissue samples were also collected and fixed in a 10% formalin buffer for subsequent routine histological staining with hematoxylin and eosin (HE). Other kidney tissue samples were collected, preserved in TissueTek OCT compound (Fisher Scientific, Pittsburgh, PA, USA) for confocal microscopy, and frozen in liquid nitrogen for mRNA isolation and quantitative real-time PCR. Serum and kidney tissue were stored at −80 °C until sample processing.

### 2.3. Histopathological Analysis

To assess the morphology and histological alterations in the brain and kidneys of infected and uninfected animals, the mice were euthanized, and the tissues were fixed in 10% formalin. After embedding the material in paraffin, tissue sections were deparaffinized in xylene for 30 min and then rehydrated in decreasing concentrations of alcohol (100%, 90%, 80%, and 70%) for 3 min each. Subsequently, slides containing the sections were immersed in Harris hematoxylin for 1 min, followed by a 10 min rinse in running water. They were then briefly immersed in eosin for 40 s and rinsed quickly in running water. For slide mounting, they were dehydrated in increasing concentrations of alcohol (70%, 80%, 90%, and 100%) for 3 min each and then subjected to a bath of xylene/alcohol (*v*/*v*) for 1 min and xylene for 15 min. The slides were mounted with a coverslip and Canada balsam [37]. A quantitative assessment of the number of microhemorrhage foci was carried out and the results were compared with the RMCBS score assigned to each animal that participated in the analysis. Midsagittal brain sections from infected mice were harvested on days 4 and 7 after PbA infection and then stained with hematoxylin and eosin (HE). HE-stained brain sections were scanned on the Apério ScanScope AT Turbo scanner (Leica, Heerbrugg, Switzerland) and the images created corresponding to the total area of the midsagittal section (mss) of each mouse were screened to detect hemorrhage foci using the Apério Image Scope software v.8.1 (Leica).

### 2.4. Morphometric Analysis of Renal Parenchyma

After the histological processing of the kidney and staining, digital photographs were taken using 10× and 40× objectives. A microscope with a DFC450 camera system (Leica, Wetzlar, Germany) was used for image capture. The morphometric analysis of hypercellularity was performed using the ImageJ image processing software v.1.53k, following the protocol described and established by others [24]. Glomerular cellularity was measured as the percentage of glomeruli showing more than 11 mesangial cells per section. At least 50 glomeruli were analyzed for each animal.

### 2.5. Serum and Tissue-Specific Cytokine Measurement

The samples of serum and kidney tissue from PbA-infected and uninfected mice were collected. Serum samples from the animals were collected and processed using the Cytometric Bead Array (CBA) Mouse Inflammation Kit according to manufacturing recommendations (BD Biosciences, San Jose, CA, USA). Tissues were homogenized and normalized to a final concentration of 200 mg/mL in PBS-EDTA 5 mM buffer containing a cocktail of protease inhibitors. The samples were centrifuged at 3000× *g* for 10 min at a temperature of 4 °C. The CBA analysis was used to quantify the cytokines IL-6, IL-10, IFN-γ, and TNF-α in the serum and kidney samples. Microspheres with distinct fluorescence intensities were conjugated with a specific capture antibody for each cytokine [38]. The reaction activity was measured using the FACSCanto II Flow Cytometer (BD Bioscience, San Jose, CA, USA). The results were analyzed, and graphs and tables were generated using the CellQuest software v.3.3 (BD Bioscience, San Jose, CA, USA).

### 2.6. Serum Measurement of Creatinine and BUN

Blood was centrifuged and plasma was obtained for the measurement of creatinine and blood urea nitrogen (BUN) using the Labtest kit (Bioclin, Belo Horizonte, Brazil) on Bioclim equipment, according to the manufacturer’s recommendations. Serum samples were centrifuged at 3500 rpm for 8 min. The samples were transferred to a tube containing EDTA and sodium citrate and centrifuged for 10 min.

The levels of creatinine and BUN were determined using a UV enzymatic reaction in a buffer containing 500 mmol/L TRIS, 5 mmol/L ADP, 100 mmol/L α-Ketoglutarate, 50 KU/L Urease, and 5 KU/L Glutamate Dehydrogenase (Bioclin, Belo Horizonte, Brazil). The samples were read at an absorbance of 340 nm.

### 2.7. Immunolabeling of IgG Agglutination and Active Caspase-3 in Renal Tissue

TissueTek OCT-embedded freezing sections (8 µm) from renal tissue were used to measure IgG deposition using direct immunofluorescence assay after cryostat sectioning. The renal tissue was incubated at room temperature for 15 min and then permeabilized in acetone at −20 °C for 15 min. An additional 15 min were used to dry the tissue at room temperature. Subsequently, the renal tissue was incubated with an Alexa Fluor 488-conjugated rabbit anti-mouse IgG antibody (1:40 diluted in 0.005% saponin, 1% BSA and PBS; A11061; Thermo Fisher Scientific, Fremont, CA, USA) for 60 min at room temperature.

For staining active caspase-3, a routine procedure was followed: the tissue was incubated for 15 min in acetone at −20 °C, followed by permeabilization with 0.1% Triton X-100 for 30 min at room temperature. The tissue was blocked with 3% BSA for 60 min at room temperature. Finally, the renal tissue was incubated with a mouse anti-active/pro-caspase-3 monoclonal antibody (1:150 diluted in 0.005% saponin, 1% BSA and PBS; 31A1067; Thermo Fisher Scientific, Fremont, CA, USA) and an Alexa Fluor 568-conjugated donkey anti-mouse IgG antibody (1:300 diluted in 0.005% saponin, 1% BSA and PBS; A10037; Thermo Fisher Scientific, Fremont, CA, USA) to immunolocalize the protein in the renal tissue [37]. Images were acquired and analyzed using the LSM510 confocal microscope (Zeiss, Thornwood, NY, USA).

### 2.8. Imunohistochemistry for Active-Pro-Caspase-3

Paraffin-embedded kidney sections were marked with the same anti-active/pro-caspase-3 antibody in order to identify apoptotic cells within the renal parenchyma of mice with 7 days of PbA-infection, as described by Parreira et al., 2009 [39]. Briefly, after blocking endogenous peroxidase for 30 min with 0.3% H_2_O_2_, sections were incubated with 10% normal goat serum for 60 min and with the mouse anti-active/pro-caspase-3 monoclonal antibody (1:50 diluted in 0.005% saponin and PBS; 31A1067) for 60 min. After washing in 50 mM Tris–HCl, sections were successively incubated with biotinylated secondary goat anti-mouse IgG L-chain biotinylated antibody (1:300 diluted in 0.005% saponin and PBS; B7151, Sigma, Burlington, MA, USA), avidin–biotin peroxidase, and diaminobenzidine (Vector Laboratories, Newark, CA, USA). Sections were viewed under a Leica DMR coupled to a Leica DC300 digital camera (Leica, Heerbrugg, Switzerland). The specificity of immunostaining was tested using incubation in the absence of primary antiserum and with control mouse IgG (Vector Laboratories). 

### 2.9. Quantitative Real-Time RT-PCR

Mouse kidney samples were homogenized in Trizol reagent (Life Technologies, Carlsbad, CA, USA) in order to extract total RNA. The total RNA was quantified by measuring the absorbance at 260 nm on a GeneQuant spectrophotometer (GE Healthcare, Chicago, IL, USA). Total RNA samples were treated with DNase I (Invitrogen, Gaithersburg, MD, USA) and reverse-transcribed into cDNA using SuperScript II RNase H reverse transcriptase (Invitrogen, Gaithersburg, MD, USA) according to Lima and colleagues [40]. Specific primers were designed using Beacon Designer 2.0 (Premier Biosoft International, Palo Alto, CA, USA; Appendix A). Real-time PCR analyses were performed in duplicate with 200 nM of both sense and antisense primers in a final volume of 25 μL using 1U of Platinum *Taq* DNA Polymerase, 2 mM MgSO4, 400 μM dNTP, and SYBR Green I (Invitrogen, Eugene, Oregon, USA) on an ABI 7500 Fast Real-Time PCR System (Thermo Fisher Scientific, Waltham, MA, USA).

The relative mRNA expression of CASP3, CASP8, BCL-2, IL-1β, IL-18, NaPi-IIa, IFN-γ, TNF-α, and TRAIL genes was investigated in adult male kidneys (n = 5), after normalization to GAPDH [Ratio = 2^ΔCt(GAPDH—Target Gene)^]. The relative changes in mRNA levels of these genes in PbA mouse kidneys were determined by comparison to the control animals after adjustment to GAPDH [41].

### 2.10. Statistical Analyses

The values were expressed as the mean ± SEM (standard error of the mean). Survival analysis of the uninfected control and PbA-infected animals was carried out using the Kaplan–Meier method. A *t*-test was used to compare means for morbidity, parasitemia, and RMCBS analysis. The means of cytokine levels between the two groups were evaluated using the Kruskal–Wallis test, followed by Dunn’s post-test. All statistical analyses were performed using GraphPad Prism 7 (GraphPad Software, San Diego, CA, USA). A *p*-value lower than 0.05 was considered statistically significant.

## 3. Results

### 3.1. PbA Infection Produced CM in Mice

In order to reproduce the phenotype of CM in mice, *Plasmodium berhgei* ANKA (PbA) parasites were used to induce clinical manifestations in C57BL/6 mice, a well-known strain susceptible to ECM. The susceptibility of mice to PbA infection was estimated using survival analysis (Figure 1A). The survival curve showed that the uninfected animals remained healthy until the end of the study, whereas the mice infected with PbA started to die on day 6 post-infection. Furthermore, the PbA-infected mice all died within 8 days of i.p., injection, except two animals that remained asymptomatic until day 12, when they were euthanized because of high parasitemia. Both asymptomatic mice were not included in the body weight, RMCBS score and parasitemia analyses. Precisely, two mice died 6 days, four died 7 days, and four died 8 days after PbA infection. The RBCs of infected mice showed parasitemia of about 11 and 12% on days 7 and 8, respectively (Figure 1B). The infected mice suffered significant mass loss caused by this intense parasitemia (Figure 1C). ECM also causes neural function impairment in mice. Therefore, we applied the RMCBS score to uninfected mice and PbA-infected mice for a quantitative assessment of their motor and cognitive functions. As shown in Figure 1D, motor and cognitive functions were preserved in the control mice but significantly harmed in the mice with CM (RMCBS score ~12) at day 7. Cognitive and motor dysfunction was strongly promoted at day 8, when infected animals showed a mean RMCBS score of approximately 8. The asymptomatic infected mice showed RMCBS scores of 16 and 17, approximately, when they were euthanized.

Motor and cognitive dysfunction in the ECM model has been widely demonstrated to be due to brain damage characterized by the accumulation of iRBCs and migrating leukocytes in brain vessels, leading to microhemorrhages. Brain sections from PbA-infected mice showed deposits of leukocytes and iRBCs in microvessels (Appendix A), a rosette-like formation (Appendix A), and extensive areas of severe microhemorrhage after 7 days of infection (Appendix A). There was no accumulation of migrating RBCs and WBCs nor any microhemorrhage focus in brain sections from uninfected mice (Appendix A). In order to establish a correlation between hemorrhage lesions in the brain of infected mice and RMCBS scores, a quantitative assessment of the number of microhemorrhage foci in midsagittal brain sections from animals after 4 and 7 days of PbA infection was performed. For each infected mouse, an RMCBS score was assigned and the results were confronted with those from the quantitative assessment of microhemorrhage foci in the brain (Appendix A). Our analysis showed a three-fold increase in the mean number of foci in mice at day 7 compared to those at day 4 post-infection. Conversely, mice at day 7 exhibited a significantly lower mean RMCBS score compared to those at day 4 (19.63 ± 0.18; n = 5), suggesting that cognitive and motor functions might not considerably decline until 6 or 7 days after PbA infection, coinciding with a substantial increase in microhemorrhage foci in brain tissue that may contribute to a reduction in RMCBS scores to approximately 12.

Taken together, these results demonstrate that our *P. berghei* ANKA infection procedure successfully provided the murine ECM model that is useful to investigate the AKI phenotype produced by malaria parasites.

### 3.2. P. berghei ANKA Infection Caused Renal Dysfunction in Mice

AKI is a disorder commonly developed by mice with CM. The measurement of plasma creatinine and BUN revealed that PbA-infected mice had significantly higher plasma levels of these two renal function parameters than uninfected animals at day 7 post-infection (Figure 2A,B). The mRNA expression level of NaPi-IIa, the major proximal tubule type II Na/Pi cotransporter, whose protein and transcript levels were demonstrated to be reduced in the kidney of rats with ischemia-induced AKI and LPS-induced endotoxaemia [42,43], was also determined. Proximal tubule cells from mice infected with the parasite exhibited about five-fold less NaPi-IIa transcripts than control animals (Figure 2C). Furthermore, the histological examination of the renal parenchyma of PbA-infected mice showed that the severe malaria condition produced morphological changes in the renal corpuscles as well as in the tubules. Kidneys from uninfected animals did not show any alteration (Figure 3A,C,E).

HE-stained kidney sections from infected animals showed the presence of migrating leukocytes in vessels (Figure 3B), an enlarged Bowman’s space, and a higher number of collapsed tufts (Figure 3D,F,G), compatible with glomerulosclerosis. Focal or diffuse tubular luminal dilatation was also observed. Loss of the brush border membrane was seen in some proximal tubules (Figure 3F), and at high magnification cytoplasmic vacuolation of tubular cells was observed as well (Figure 3H). It was noticed that several nuclei displaced into the tubular lumen, suggesting the detachment of cells from the basement membrane (Figure 3H). Moreover, a mild segmental and focal increase in mesangial cells in PbA-infected mice was detected (Figure 4A). This augmentation of the mesangial cell number indicates hypercellularity. In addition, infected mice showed glomeruli with sclerotic areas (Figure 4B). Taken together, these results demonstrate that the murine malaria parasite *P. berghei* ANKA induces AKI during ECM.

These findings from the ECM model used in our study reinforce the typical renal phenotype of AKI demonstrated in previous studies using the same model.

### 3.3. Serum Cytokine Levels Were Increased in Mice with CM

Cytokine- and cell-mediated immune responses to *Plasmodium* infection are thought to play important roles in the pathogenesis of associated glomerulopathies in humans and mice. Since it has been shown that blood levels of cytokines are altered during malaria, levels of the serum cytokines TNF-α, IFN-γ, IL-10, and IL-6 of uninfected and PbA-infected mice at day 7 post-infection were determined using CBA analysis (Appendix A). PbA-infected mice showed significantly higher serum levels of all four investigated cytokines IFN-γ, TNF-α, IL-10, and IL-6 when compared to control mice. These results point to an involvement of these circulating cytokines in the immune-related pathogenesis of AKI caused by *P. berghei* ANKA infection in mice.

### 3.4. PbA-Infected Mouse Kidneys Display Glomerular IgG Deposition 

Some studies have suggested an essential role of glomerular deposition of immune complexes in the pathophysiology of malaria-associated nephritis. Therefore, we decided to investigate the formation of glomerular IgG deposits in mice with CM. A direct immunofluorescence assay showed IgG deposits in the glomerulus of *P. berghei* ANKA-infected mice after 7 days of infection, and only a very faint autofluorescence signal was observed in uninfected mice (Figure 5A–C). At high magnification, the analysis of IgG distribution in Bowman’s capsule space and glomerular area of PbA-infected mice showed strong deposition of this immunoglobulin (Figure 5D). These findings indicate a relationship between IgG deposition in the glomerulus of infected mice and the AKI produced in the *P. berghei* ANKA infection model used in the present study.

### 3.5. Kidney Tissue from ECM Mice Showed Potent Inflammatory Response

Researchers have made recent efforts to establish the effects of pro- and anti-inflammatory cytokines on different cellular processes that lead to the onset of glomerulopathies and tubular lesions caused by malaria parasites. To determine the mRNA expression levels of IFN-γ, TNF-α, IL-1β, and IL-18, as well as the abundance of cytokines IFN-γ, TNF-α, IL-6, and IL-10 in the kidneys of uninfected and PbA-infected mice at day 7 post-infection, we performed quantitative real-time PCR and CBA assays, respectively. Infection with this malaria parasite promoted the upregulation of genes encoding all the cytokine mRNAs quantified in the whole mouse kidney (Figure 6A–D). In addition, the CBA assay revealed that *P. berghei* ANKA infection induced an increased concentration of the pro-inflammatory cytokines IFN-γ, TNF-α, and IL-6 in the kidney of infected mice (Figure 7A–D). Similarly, the abundance of the anti-inflammatory cytokine IL-10 in the kidney of PbA-infected mice was higher than that of uninfected control animals.

These results suggest an important role for these cytokines in both glomerulopathy and nephrotic syndrome caused by *P. berghei* ANKA, which may be associated with mesangial and tubular apoptosis.

### 3.6. Apoptosis-Related Proteins Were Upregulated in the Renal Tissue of Mice with CM

To investigate the association of apoptosis pathways with the mouse renal phenotype generated by *P. berghei* ANKA after 7 days of infection, the mRNA expression levels of TRIAL, BCL-2, CASP3, and 8 in the renal tissue of uninfected and PbA-infected mice were determined using quantitative real-time PCR analysis. In addition, in situ immunofluorescence assays were also carried out on kidney samples from uninfected control and CM animals to identify structures of the renal parenchyma and cells affected by the apoptotic process. A quantitative RT-PCR assay revealed that PbA infection upregulated TRAIL mRNA in mouse kidneys (Figure 8A). In contrast to uninfected mice, the renal expression of TRIAL mRNA was about 12-fold higher in infected mice. Moreover, ECM mice on day 7 of infection showed a two-fold increase in the mRNA expression of CASP3 and 8 (Figure 8B,C). On the other hand, no significant difference in the mRNA expression of BCL-2 was observed between infected and uninfected animal groups (Appendix A). From in situ immunofluorescence analysis, we observed that the antibody against active/pro-caspase 3 produced only a weak signal in kidney sections of uninfected mice (Figure 9A,C,E,G), whereas in PbA-infected mice it generated an intense signal in some renal tubules and glomeruli (Figure 9B,D,F,H).

The immunohistochemical analysis of renal tissue sections from mice with seven days of infection showed intense staining for CASP3 in tubular and renal corpuscle cells. In the cortical region, glomeruli with varying degrees of collapse exhibited a signal, which was observed in podocytes and mesangial cells with a predominance in the latter (Figure 9I). Conversely, intact glomeruli did not reveal staining. Furthermore, both proximal and distal convoluted tubules exhibited strong cytoplasmic staining (Figure 9J). Collecting ducts in the medullary zone also displayed strong staining for CASP3, with no discrimination between principal and intercalated cells (Figure 9K).

Since the differentially expressed TRIAL, CASP3, and 8 mRNAs encode mediators that play central roles in TRAIL apoptosis, our results indicate that they may contribute to the pathogenesis of AKI observed in ECM.

## 4. Discussion

AKI is a clinical manifestation widely recognized to be shared by ECM and severe human malaria. In this study, we present relevant findings on malaria-associated AKI, wherein the expression levels of cytokines and apoptosis mediators in the renal tissue and blood of mice with severe malaria were markedly altered during the blood stage parasite-induced infection. Studies have demonstrated that the renal function of mice progressively deteriorates throughout the course of PbA parasitemia [18,29,44,45]. Serum levels of BUN and creatinine represent useful tools for investigating the decline in renal function in this malaria animal model. Consistently, our results confirmed that mice severely affected by malaria associated with PbA had impaired renal function, as their serum levels of creatinine and BUN on the seventh day of infection were significantly higher compared to healthy animals. Elevated serum levels of BUN and creatinine are a common feature in mice with severe malaria, although this hallmark is not restricted to animals susceptible to CM. For instance, the malaria-resistant murine model, in which Balb/c mice are infected with PbA, exhibited AKI with substantially elevated serum levels of BUN and creatinine [29,44]. Similarly, susceptible C57BL/6 mice infected with *P. berghei* NK65, a parasite strain that does not cause severe malaria in rodents, also showed elevated blood levels of these two renal function biomarkers [13]. Other *Plasmodium* species, such as *P. yoelii*, also induced the same phenomenon in mice [24,46]. Our findings with serum BUN and creatinine reinforce the results found by other authors who worked with susceptible and resistant mouse models of malaria [18,29,44,45].

Abnormal serum phosphate concentrations frequently manifest in critically ill individuals and patients with severe malaria often exhibit hypophosphatemia. In fact, hypophosphatemia is observed in around 40% of patients with acute falciparum malaria [7,47]. Nevertheless, it is not determined whether alterations in renal phosphate handling are influenced by a factor that directly correlates with serum Pi concentrations. From our data, a marked downregulation in gene expression of the sodium/phosphate tubular co-transporter NaPi-IIa was observed in the kidney of ECM mice. To our knowledge, this is the first time that a change in the renal expression of NaPi-IIa transcripts is reported in mice with malaria-associated AKI. NaPi-IIa seems to be the key player in determining brush-border membrane sodium/Pi co-transport and reabsorption, being responsible for about 70% of the renal phosphate transport [48]. Interestingly, the disruption of the *Npt2a* gene in mice resulted in hypophosphatemia and phosphaturia [49]. Previous studies in mouse and rat models of renal failure, such as ischemia-reperfusion injury, anti-Thy1.1 glomerulonephritis (GN) and IgG-induced mesangioproliferative GN, have shown a notable decrease in NaPi-IIa expression in proximal tubular cells [42,50,51]. Despite the fact that phosphate clearance is not examined in our ECM model, it is reasonable to assume that renal downregulation of NaPi-IIa could lead to hypophosphatemia in ECM mice, similar to what may occur in an expressive proportion of patients that develop CM and hypophosphatemia. Therefore, we propose that a partial impairment of proximal tubular Pi reabsorption, resulting from reduced expression of NaPi-IIa mRNA, may be one of several factors contributing to severe hypophosphatemia caused by *P. falciparum* infection.

There is convincing evidence linking high serum levels of inflammatory cytokines, such as IL-6, IL-1β and TNF-α, with hypophosphatemia in patients with sepsis [52]. Likewise, serum IL-6 and TNF-α levels were also found to increase in our model when mice were infected with PbA. Accumulated studies suggest the involvement of the NF-κB transcription factor in the pathogenesis of AKI underlying IgA nephropathy and inflammation [53]. Furthermore, NF-κB is a central mediator of signal transduction stimulated by several major inflammatory cytokines, such as TNF-α and IL-1, thereby participating in the effector phase of inflammation [54]. For example, Han and colleagues (2022) demonstrated that a consistent downregulation of the *Oct2* gene in the kidney of rats with AKI was associated with TNF-α upregulation in an NF-κB-dependent manner [55]. *Oct2* is the gene that encodes for the organic cation transporter 2 in many tissues, including the kidney. While these cytokines might indirectly modulate NF-κB target genes, whose expression is altered in individuals with severe infections, it is still far from being elucidated whether *Npt2a* downregulation incited by cytokine-related transcription factors truly contributes to hypophosphatemia in malaria. Currently, our group is working towards better understanding the molecular mechanisms involved in the transcriptional impairment of the *Npt2a* gene in mice with CM.

Malaria-associated AKI in mice has been attributed to the development of glomerular lesions, such as epithelial aggregates, basement membrane thickening, and the proliferation and thickening of mesangial cells [26]. It is known that a pronounced decrease in the glomerular filtration rate (GFR) can lead to the emergence of these lesions due to a reduction in flow and blood pressure in the glomerular capillaries. Therefore, it would not be unreasonable to support the idea that the obstruction of glomerular capillaries by IgG immune complexes is responsible for the reduction in GFR and, consequently, the glomerular lesions observed in animals infected with malaria parasites [44]. Our histological examinations of brain and kidney sections from infected mice showed leukocyte migration and iRBC sequestration in blood vessels. The sequestration of iRBCs and the recruitment of immune cells in the glomerular capillaries are assumed to be essential factors for the deposition of IgG immune complexes. IgG aggregates, in turn, induce endothelial activation, hemodynamic imbalance, mesangial cell proliferation and sclerosis, apoptosis and tubular necroses contributing to the pathogenesis of malaria-associated AKI [23,24,25]. In addition to the cytoadherence of iRBCs and chemotaxis of mononuclear immune cells, a strong local expression of pro-inflammatory cytokines such as TNF-α, IL-1β and IL-6 seems to be a critical event in the establishment of tubulointerstitial injury and mesangioproliferative glomerulonephritis during murine severe malaria [19,28]. We were able to show an exacerbated stimulation of TNF-α and IL-6 protein production in the whole kidney in response to PbA infection. A twofold increase in IL-1β mRNA was demonstrated as well. In light of these findings, we consider that these results provide additional support for the hypothesis that pro-inflammatory cytokines expressed locally in the kidney may play a role in the pathogenesis of AKI associated with ECM. In contrast, a significant augmentation of IL-10, a potent anti-inflammatory interleukin, was also observed. A previous study definitively demonstrated that IL-10 plays a protective role in ECM caused by PbA [56]. Moreover, it is believed that the production rate of IL-10 and both IFN-γ and TNF-α by human CD4^+^ T cells, and the regulated shift from one to the other during the course of infection, are crucial for the development of an appropriate but self-limiting immune response [57]. Whether this induction of IL-10 exerts any inhibitory effect on the AKI-inducing inflammatory response triggered by IL-6 and TNF-α is a matter that was not addressed in our study, but it is a possibility that should be taken into account. 

A quantitative analysis of the renal levels of IL-10, IFN-γ and TNF-α cytokines at different time points revealed an inverse relationship between IL-10 and the other two cytokines over the course of PbA infection in mice [29]. The authors showed that while IL-10 levels decreased, IFN-γ and TNF-α levels increased, diminishing the IL-10 production rate relative to the other two cytokines, thus favoring a pro-inflammatory effect at the late stage of infection. This effect is presumably responsible for the ongoing loss of cytoprotection, leading to renal damage. Moreover, Singh and colleagues demonstrated that IL-18 plays a protective role in host defense by enhancing IFN-γ production during murine malaria blood-stage infection [58]. We observed a robust increase in the mRNA and protein expression of IFN-γ and TNF-α in the kidney of mice with CM. Further, our CBA analyses highlighted a very low renal induction of IL-10 in comparison to the robust induction of IFN-γ and TNF-α in mice after seven days of infection. Therefore, the elevated IFN-γ could further contribute to undermining the anti-inflammatory effect of IL-10 in ECM. As we did not monitor IL-10 production throughout the evolution of PbA infection, we can only speculate that there might be a stronger induction of IL-10 at the early stage, gradually decreasing until its low levels exert a weak counteracting effect, which could explain the pronounced inflammatory response intensified at the late stage of ECM. 

The histopathological changes we observed in glomeruli and proximal tubules of mice with ECM comprise glomerular collapse, mesangial hypercellularity, the loss of brush border, cytoplasmic vacuolization and the nuclear detachment of tubular epithelial cells. These tissue alterations are commonly found in mice and patients suffering from severe malaria, who develop glomerulonephritis, acute tubular necrosis and interstitial nephritis [14,27]. Immunohistochemistry analysis showed that TNF-α, IL-6 and IL-10 were mainly expressed by mesangial cells and infiltrating macrophages in the glomeruli, as well as in the tubulointerstitial area and endothelium during the course of tubulointerstitial and glomerular nephritis in ECM [19,28]. Furthermore, mesangial cell proliferation and mesangial matrix accumulation are common features in mesangioproliferative glomerulonephritis (MPGN), a distinct glomerular response pattern characterized by diffuse or focal increase in the number of mesangial cells and expansion of the extracellular matrix (fibrosis) in the glomerular mesangium with or without immunoglobulin deposition [59]. We consider two previously presented hypotheses [19], aiming to explain a highly probable paracrine and autocrine action of these cytokines during the pathogenesis of severe malaria-associated MGPN: the first one suggests that infiltrated macrophages release INF-γ, which, in turn, modulates the production of TNF-α, IL-6 and IL-10 by activated mesangial cells. The second proposes that the deposition of IgG aggregates in glomerular tufts induces mesangial cells to synthesize these cytokines. From our data, there was IgG deposition, leukocyte migration and renal induction of the three above-mentioned cytokines plus IFN-γ. It is conceivable that the intrinsic glomerular cells and infiltrating inflammatory cells may engage in crosstalk during the infection, allowing endogenous proinflammatory cytokines to modulate local immune reactions in the glomerulus.

The heightened synthesis of IL-6 in the glomerulus may act to stimulate the local proliferation and differentiation of mesangial cells, thereby contributing to the complex processes leading to fibrosis and glomerulosclerosis. Similarly to IgG deposition in ECM, the deposition of anti-Thy1.1 antibodies in the glomerulus of rats with MPGN stimulated mesangial cells to produce IL-6. Interestingly, the silencing of the *c/ebp* gene, which encodes the transcription factor C/EBP1 that binds to the IL-6 gene promoter and positively controls its expression in synoviocytes, led to the downregulation of IL-6 and ameliorated mesangial cell proliferation and extracellular matrix expansion in rats with Thy1.1 nephropathy [60]. In addition, profound mesangial cell proliferation was also observed in the kidney of IL-6 transgenic mice, displaying a typical pathology of MPGN [61]. Altogether, these results directly implicate immunoglobulin deposition in mesangial cell proliferation and glomerular fibrosis by prompting mesangial cells to increase the IL-6 production.

Experimental evidence supports a pathogenic role for tubular and glomerular apoptosis in AKI [32,62]. In normal human kidneys, the rate of glomerular cell apoptosis is typically low, but it can significantly rise in glomerulonephritis, such as in acute post-infectious glomerulonephritis. Apoptosis may play a role promoting recovery in post-infectious glomerulonephritis associated with falciparum malaria [27]. MPGN with mesangial hypercellularity shows potential reversibility with appropriate treatment, indicating that excess mesangial cells can be successfully eliminated through apoptosis. Studies performed in the Thy1.1 nephritis model, demonstrating the complete resolution of mesangial hypercellularity, strongly suggested that apoptosis is the major cell clearance mechanism counterbalancing cell division, thereby mediating the resolution of glomerular hypercellularity [63,64]. Consistent with these discoveries, we observed a marked increase in the renal expression of CASP3, CASP8 and particularly TRIAL mRNAs in the kidney tissue of mice with malaria-associated AKI. To date, this is the first report linking the TRAIL apoptosis pathway to AKI in mice with ECM. In addition, our in situ immunofluorescence assay revealed an active CASP3 signal in glomeruli and proximal tubules of PbA-infected mice. A recent study also found strong CASP3 labeling of these nephron segments in kidney biopsies from patients with severe malaria [31]. On the other hand, BCL-2 mRNA levels did not differ between infected and uninfected control mice, suggesting that the mitochondrial phase of apoptosis, which partially requires BCL-2 to be activated, may not be implicated. Apoptosis plays a pivotal role in kidney diseases, with the intricate involvement of both intrinsic (mitochondrial) and extrinsic (death receptor-initiated) pathways contributing to renal cell demise. The extrinsic pathway is triggered by external signals, typically involving the binding of death ligands to death receptors on the cell surface. TRAIL is one such death ligand that can promote apoptosis by binding its receptors, leading to the downstream activation of CASP3 and 8 and subsequent cell death. The involvement of TRIAL in renal injury and apoptosis in kidney diseases, such as ischemia reperfusion injury (IRI), lupus nephritis, minimal-change nephrotic syndrome, chronic kidney disease and polycystic kidney disease, has been documented [33,34]. Adachi and collaborators (2013) showed that TRAIL blockade inhibited tubular apoptosis, reduced the accumulation of neutrophils and macrophages, and attenuated renal fibrosis and atrophy in mice after IRI [34]. However, the data strongly suggested that the upregulation of TRAIL in tubules from patients with proliferative lupus nephritis may play a protective role by enhancing proximal tubular epithelial cell survival, although exerting a proinflammatory effect that may contribute to local inflammation and injury [33]. 

Currently, the available literature data regarding the interaction between parasites and the host regarding the stimulation of the TRAIL-dependent apoptosis pathway is lacking. Although a close relationship between TRAIL expression in hosts and parasitic infections may still seem uncommon, it has been demonstrated that the induction of TRAIL in certain parasite diseases is crucial for host defense, modulating responsive apoptosis processes [65,66]. For instance, the kidney-derived Vero cell line highly expressed TRIAL, CASP3 and TNF-α in response to the Apicomplexan parasite *Toxoplasma gondii* infection [65]. The authors also observed that the early apoptotic cells infected with *T. gondii* tachyzoites increased significantly compared with control cells, suggesting that the TRAIL pathway is implicated in the *T. gondii*-induced apoptosis of this kidney cell line. Moreover, it has also been reported that *Leishmania major*-infected primary human epidermal keratinocytes abundantly express TRAIL and IFN-γ proteins [66]. Importantly, this study showed that when using TRAIL-containing supernatants from *L. major*-infected PBMC cultures, neutralizing TRAIL by adding an anti-TRAIL monoclonal antibody before incubation with HaCaT cells blocked up to 75% of apoptosis. It is likely that TRIAL modulates apoptosis, attempting to attenuate parasitemia by reducing the number of parasitic cells and thus ameliorating tissue damage. Similarly, parasite-activated mesangial cells may likely express TRAIL, CASP3 and CASP8 to resolve glomerular hypercellularity by triggering apoptosis and consequently ameliorating malaria-induced glomerulonephritis in ECM.

## 5. Conclusions

In conclusion, our study sheds light on crucial aspects of severe malaria-associated AKI, particularly in the context of ECM. The observed alterations in serum biomarkers, including elevated creatinine and BUN levels, underscore the impairment of renal function during the progression of parasitemia. Notably, our investigation revealed a marked downregulation of the sodium/phosphate tubular co-transporter NaPi-IIa in the kidney of PbA-infecetd mice, providing novel insights into the pathogenesis of hypophosphatemia in CM. We propose that this downregulation, potentially influenced by proinflammatory cytokines, contributes to the observed hypophosphatemia, although further mechanistic studies are warranted. Furthermore, the study highlights the intricate involvement of proinflammatory cytokines, such as TNF-α, IL-1β and IL-6, in the renal pathogenesis of ECM. The observed imbalance between these proinflammatory cytokines and the anti-inflammatory cytokine IL-10, especially at later stages of infection, suggests a potential role in the perpetuation of inflammation and subsequent renal damage. Additionally, our findings unveil the significant upregulation of the TRAIL apoptosis pathway in the renal tissue of ECM mice, implicating this pathway in the mediation of apoptosis in both glomerular and tubular compartments. While our study provides novel insights into the renal alterations associated with severe malaria, further research is needed to decipher the intricate molecular mechanisms and potential therapeutic interventions in malaria-associated AKI.

## Figures and Tables

**Figure 1 pathogens-13-00376-f001:**
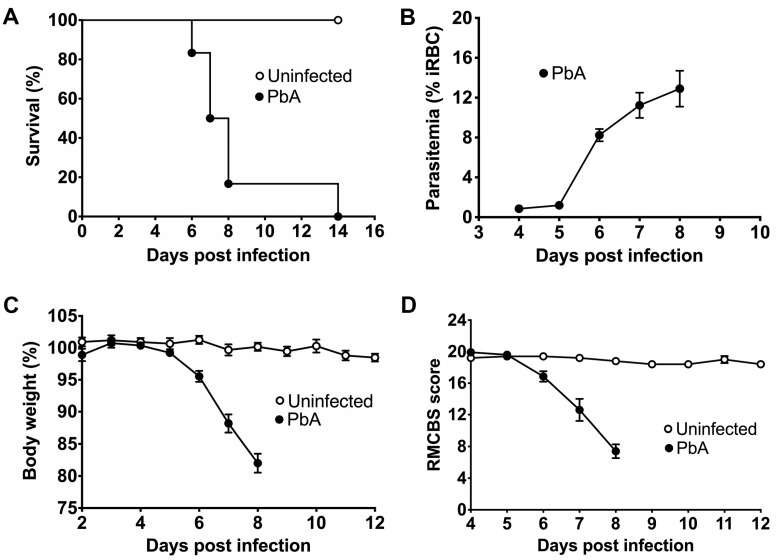
Survival analysis, parasitemia, body weight measurement, and RMCBS score estimation. Male C57BL/6 mice were intraperitoneally inoculated with 10^5^ infected red blood cells with *P. berghei* ANKA and saline solution. (**A**) Survival curves of uninfected control (n = 5) and PbA-infected mice (n = 12) estimated using the non-parametric Kaplan–Meier method. Parasite-infected mice die from 6 to 8 days after injection, while all control mice remain healthy. (**B**) Parasitemia levels (ring to schizont forms in iRBC) for infected mice (n = 10) at different time points of infection using thick and thin blood smears stained with panoptic. The percentage of iRBC significantly increases from 5 days post-infection, reaching about 13% at day 8. (**C**) Body weight measurement (%) for uninfected (n = 5) and PbA-infected mice (n = 10) at different time points. Uninfected mice do not experience weight loss, but *P. berghei* reduces the body mass of infected animals by approximately 20% within 8 days. (**D**) Estimation of RMCBS scores for control (n = 5) and PbA-infected mice (n = 10). Control mice did not show any significant change in RMCBS scores around 20 during the experiments. In contrast, infected mice showed decreased RMCBS scores from day 6 post-infection (day 6, 16.84 ± 0.82, mean ± SEM; day 7, 12.4 ± 1.41; and day 8, 7.90 ± 0.89). The cognitive and motor functions of parasite-free mice are maintained throughout the experiment, whereas in infected mice, the RMCBS score drastically decreases to 7.4, suggesting neural function impairment. All control uninfected mice were euthanized 24 days after initiating the experiments, while infected animals were maintained alive until death.

**Figure 2 pathogens-13-00376-f002:**
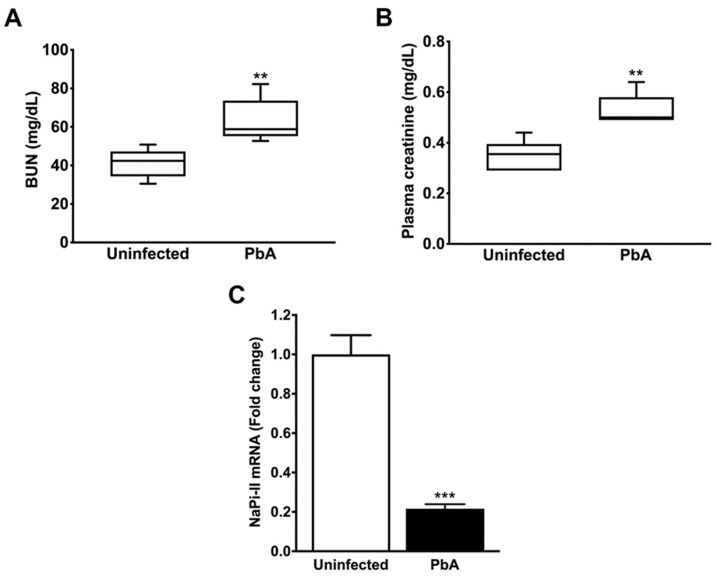
Mouse renal function evaluation at day 7 post-infection. (**A**) Biochemistry analysis showed that PbA-infected animals (n = 6) exhibited significantly higher levels of BUN (41.12 ± 3.34 mg/dL; ** *p* = 0.007) as compared to the uninfected group (n = 6; 63.30 ± 5.12 mg/dL). As seen, *P. berghei* ANKA causes reduced urinary urea nitrogen excretion in mice with ECM. (**B**) Serum creatinine levels were measured in uninfected (n = 6) and PbA-infected animals (n = 6), showing values of 0.35 ± 0.02 mg/dL and 0.55 ± 0.03 mg/dL, respectively, (** *p* = 0.000). These data demonstrate that the malaria parasite promotes a decrease in creatinine clearance in infected mice. (**B**,**C**) The RMCBS scores for control and PbA-infected mice were 19.20 ± 0.20 and 11.33 ± 0.33, respectively. (**C**) Quantitative RT-PCR analysis shows that NaPi-II mRNA expression was strongly downregulated in the kidneys of PbA-infected animals (0.22 ± 0.02 fold change; *** *p* = 0.0001; n = 5) in contrast to controls (1.01 ± 0.10 fold change; n = 5). Reduced NaPi-II transporter abundance in the proximal tubule cell membrane indicates hypophosphatemia in mice with malaria-induced AKI. Serum samples were collected and tissue samples were harvested from mice euthanized at day 7 post-infection. (**C**) The RMCBS scores for control and PbA-infected mice were 19.41 ± 0.25 and 11.60 ± 0.25, respectively.

**Figure 3 pathogens-13-00376-f003:**
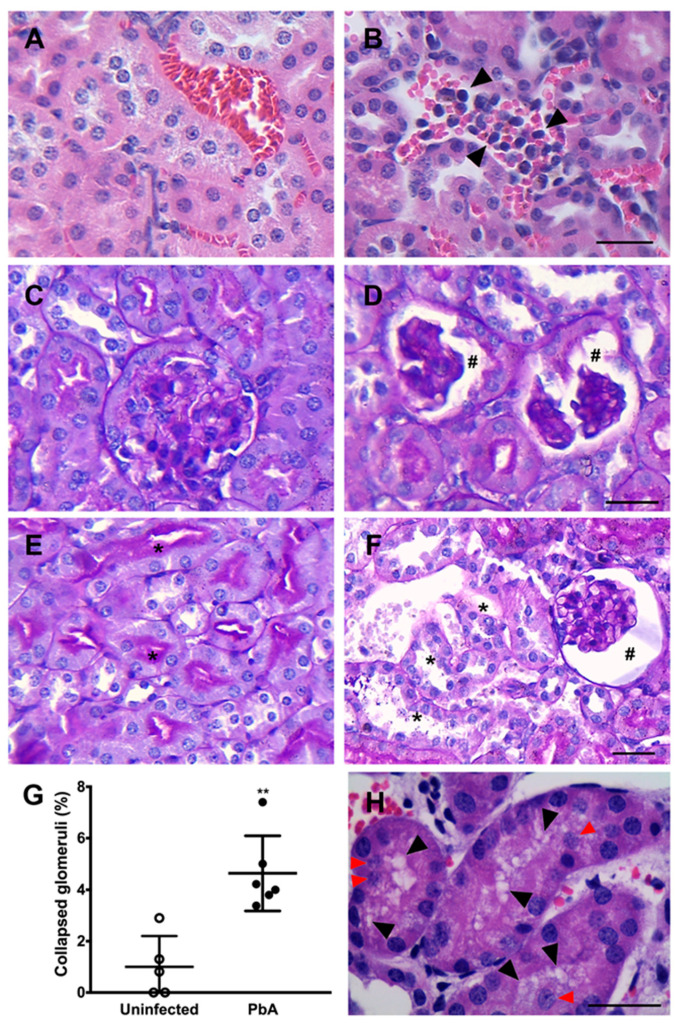
Histopathological changes in the renal parenchyma. (**A**) HE-stained kidney section depicting the renal parenchyma of an uninfected animal with unchanged renal tubules and blood vessels. (**B**) On the other hand, leukocytes and iRBC are observed within the blood vessels of infected mice in this micrograph (arrow heads). Consistent with the presence of these cell types in the kidney blood vessels of PbA-infected mice, leukocyte migration and iRBC sequestration might be occurring. (**C**) PAS-stained kidney section from an uninfected mouse showing a normal renal corpuscle. (**D**) In contrast, collapsed glomerular tufts and enlarged Bowman’s space of infected mice are observed (hashtag symbols). These renal histopathological features are suggestive of collapsing focal segmental glomerulosclerosis (FSGS) or a less severe and/or earlier form of AKI. (**E**) Proximal tubular cells exhibiting robust brush border signal (asterisk). (**F**) Conversely, loss of brush border is clearly noticed in some convoluted proximal tubules of PbA-infected mice (asterisk); a collapsed glomerular capillary tuft is also observed (hashtag). (**G**) Morphometric analysis of whole sections showing the percentage of collapsed glomeruli in the renal parenchyma of uninfected (n = 5) and PbA-infected animals (n = 6), and revealing 1.0% and 4.6% of collapsed glomeruli, respectively (** *p* = 0.002). The RMCBS scores for control uninfected and infected mice were 19.20 ± 0.45 and 11.33 ± 0.33, respectively. These data corroborate the histopathological findings. (**H**) Image showing cytoplasmic vacuolization of proximal tubular epithelial cells (black arrow heads) and nuclear detachment (red arrow heads) in infected mice. Cytoplasmic vacuolization and nuclear detachment are typically events that occur in kidneys with AKI. Tissue samples were harvested from mice euthanized at day 7 post-infection. (**A**–**F**), original magnification: ×40. (**H**): ×100. Scale bars: 40 μm.

**Figure 4 pathogens-13-00376-f004:**
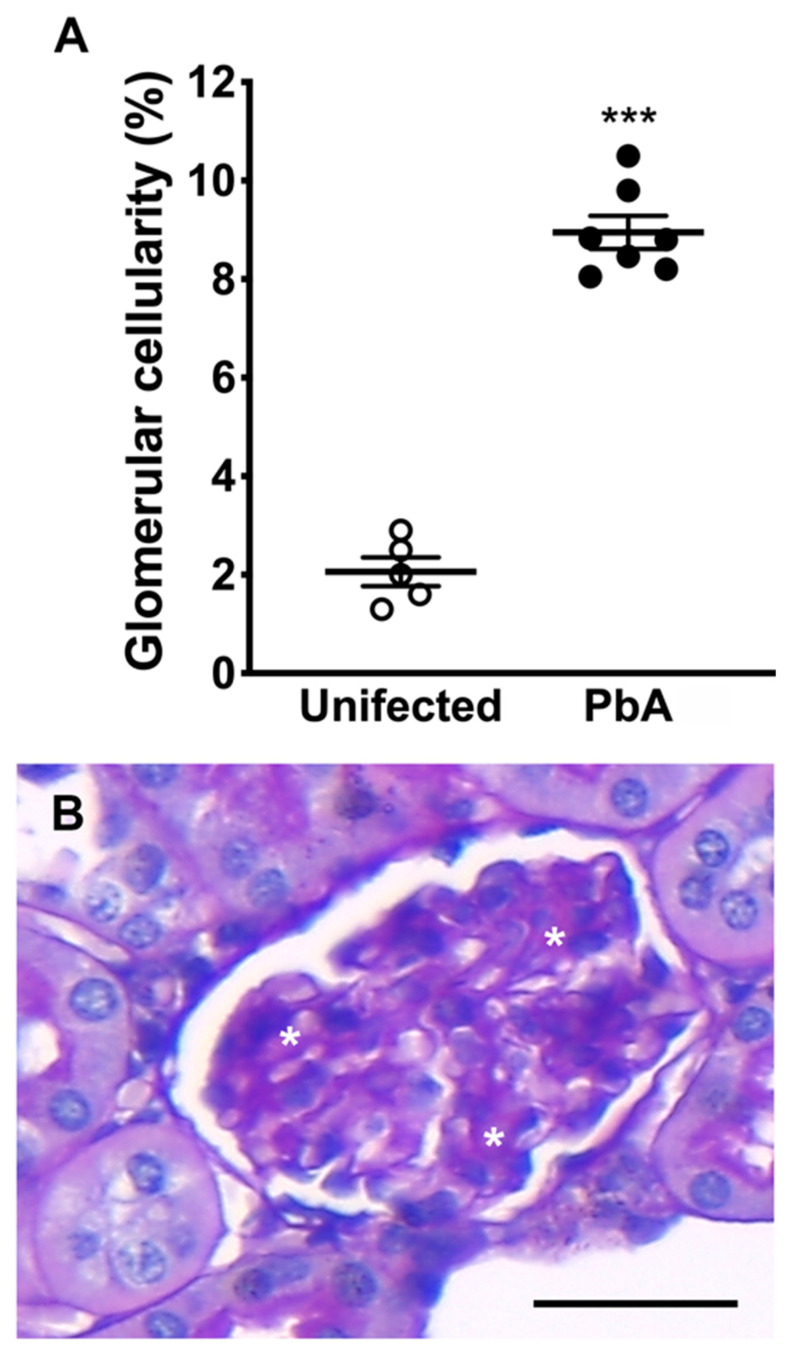
Morphometric examination of glomerular hypercellularity and sclerosis. (**A**) Morphometric analysis of whole kidney sections. Results reveal a significantly higher percentage of glomeruli containing a greater number of mesangial cells in ECM mice. Hypercellularity is observed in 2.6% of glomeruli of uninfected (n = 5) in comparison to 8.9% of PbA-infected mice (*** *p* = 0.000; n = 7). The RMCBS scores for control uninfected and infected mice were 19.00 ± 0.32 and 11.57 ± 0.30, respectively. (**B**) Sclerotic areas (asterisks) in a glomerulus of a *P. berghei* ANKA-infected mouse. Glomerulosclerosis and mesangial hypercellularity are two key features of MPGN. Tissue samples were harvested from mice euthanized at day 7 post-infection. Original magnification: ×40. Scale bar: 40 μm.

**Figure 5 pathogens-13-00376-f005:**
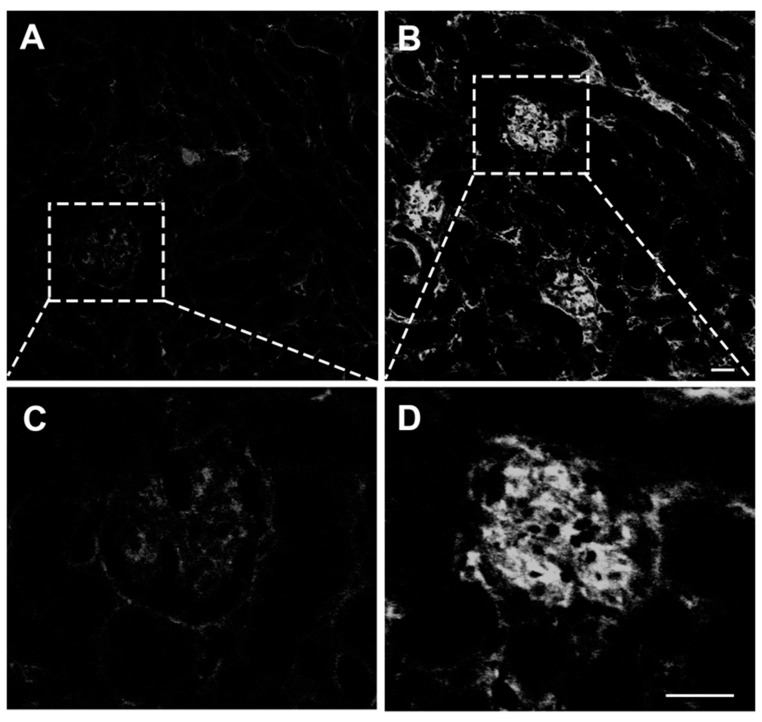
Direct immunofluorescence assay of IgG-containing immune complexes in glomeruli. (**A**,**C**) Confocal microscopy of cryostat sections of control kidneys labeled with an immunofluorescent anti-IgG antibody by using direct labeling approach revealed only a very faint auto-fluorescent signal. (**B**,**D**) By contrast, the glomeruli of animals infected with the parasite exhibited intense labeling. These histological findings demonstrate the deposition of immune complexes, at least, in the glomerular capillaries of infected animals, which is thought to be essential for the pathogenesis of acute immune complex glomerulonephritis in mice with ECM. Tissue samples were harvested from mice euthanized at day 7 post-infection. (**A**–**D**), original magnification: ×63. Scale: 40 µm.

**Figure 6 pathogens-13-00376-f006:**
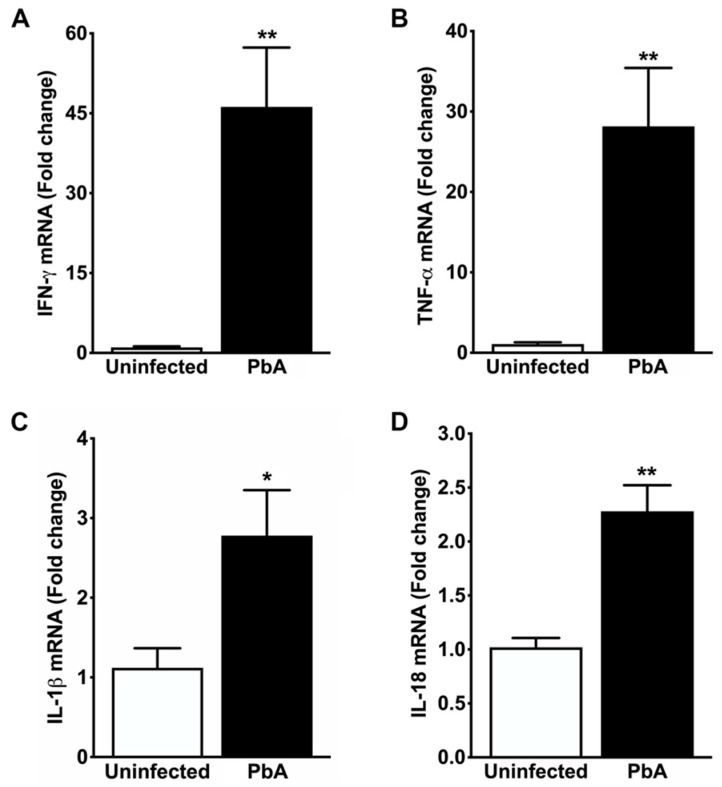
Cytokine gene expression in the whole kidney. (**A**) Real-time RT-PCR analysis shows exaggerated upregulation of IFN-γ mRNA in the kidney of mice seven days after infection with PbA (46.20 ± 11.13 fold change; ** *p* = 0.004) when compared to uninfected controls (1.04 ± 0.19 fold change). (**B**) Real-time RT-PCR analysis also shows strong overexpression of TNF-α mRNA in PbA-infected mice (28.16 ± 7.26 fold change; ** *p* = 0.006) in contrast to controls (1.08 ± 0.22 fold change). This remarked induction of IFN-γ and TNF-α transcripts in the renal tissue of ECM mice suggests increased parasite sequestration in the endothelium and recruitment of activated leukocytes to the kidney, in addition to an intense acute local inflammatory response associated with glomerular and tubule-interstitial injuries. (**C**) Real-time RT-PCR analysis demonstrates that IL-1β gene expression is about three-fold higher in PbA-infected mice (2.78 ± 0.57; * *p* = 0.028) than in uninfected animals (1.12 ± 0.25 fold change). These results suggest that IL-1β, whose release from PbA-infected splenocytes is mediated by CASP8, is involved in kidney injury during ECM pathogenesis. (**D**) Real-time RT-PCR analysis reveals that IL-18 gene expression is significantly induced in PbA (2.28 ± 0.24; ** *p* = 0.001) as compared with uninfected mice (1.02 ± 0.09). This induction suggests an early protective effect of IL-18 against PbA infection by increasing IFN-γ in kidneys. N = 5 for uninfected controls and PbA-infected mice. The RMCBS scores for these animals were 19.41 ± 0.25 and 11.60 ± 0.25, respectively. All animals were euthanized on day 7 post-infection and tissue samples were harvested.

**Figure 7 pathogens-13-00376-f007:**
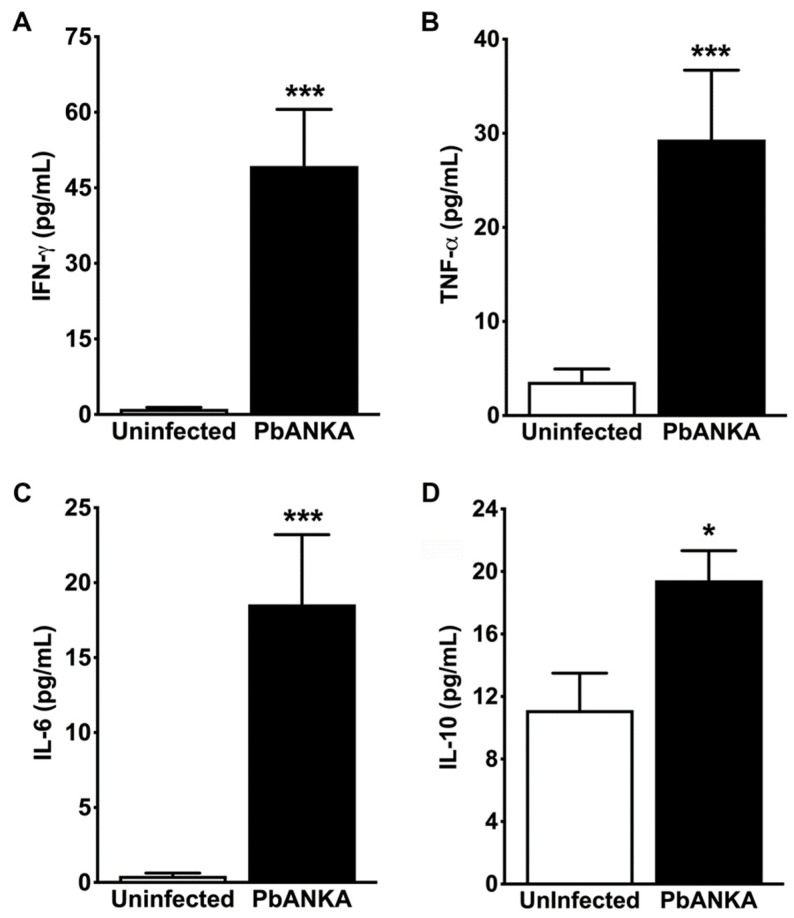
CBA analysis of cytokines in renal tissue. (**A**) Cytokine measurement shows a strong induction of IFN-γ in the renal parenchyma of PbA-infected mice (PbA, 49.32 ± 11.24 mg/dL; *** *p* = 0.006) as compared with uninfected controls (1.10 ± 0.31 mg/dL) after 7 days of inoculation. (**B**) Cytokine measurement also shows that TNF-α concentration in the renal tissue was much higher in infected mice (29.32 ± 7.37 mg/dL; *** *p* = 0.001) than in uninfected controls (3.58 ± 1.36 mg/dL). These results are in agreement with those from real-time RT-PCR analysis for both IFN-γ and TNF-α mRNA in kidneys of infected and uninfected mice at the same time point. (**C**) CBA analysis reveals that *P. berghei* infection enhances IL-6 production in the kidney parenchyma of infected mice (18.55 ± 4.65 mg/dL; *** *p* = 0.001) in contrast with uninfected controls (0.44 ± 0.18 mg/dL). (**D**) Using the same approach, the study also shows significantly higher levels of IL-10 in kidney tissue from PbA-infected mice (19.43 ± 1.90 mg/dL; * *p* = 0.017) than in control animals (11.13 ± 2.36 mg/dL). Higher levels of IL-6 and IL-10 in renal tissue from PbA-infected mice indicate that it may be involved in malaria-associated AKI, since AKI is characterized by an increase in pro-inflammatory cytokines, including IFN-γ, TNF-α, and IL-6, as well as the anti-inflammatory cytokine IL-10, in renal cortex segments. N = 5 and n = 7 for uninfected controls and PbA-infected mice, respectively. The RMCBS scores for these animals were 19.00 ± 0.32 and 11.57 ± 0.30, respectively. All animals were euthanized on day 7 post-infection and serum samples were collected.

**Figure 8 pathogens-13-00376-f008:**
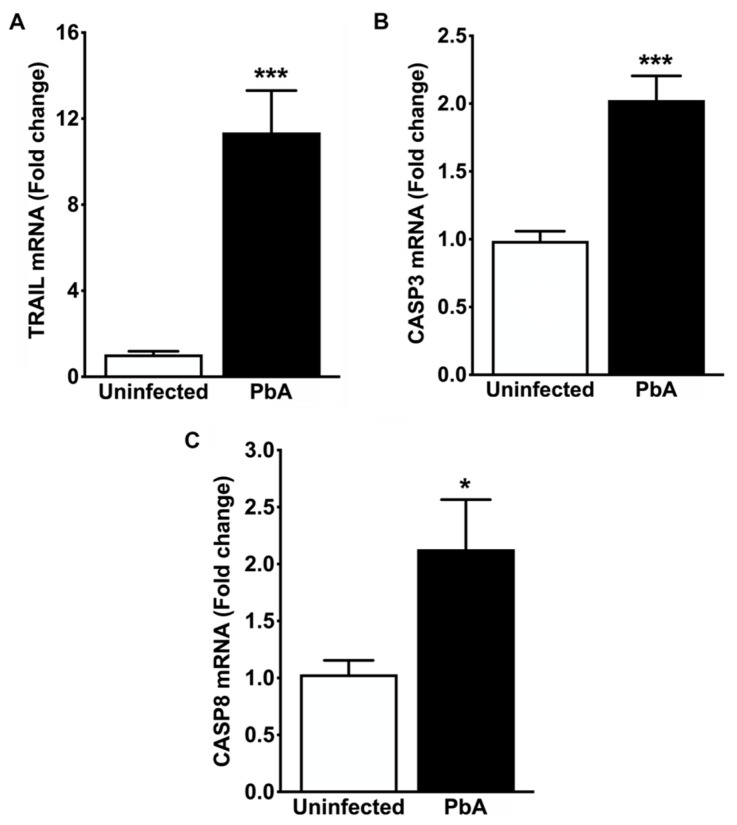
Apoptosis-related gene expression in the whole kidney. (**A**) Real-time RT-PCR analysis reveals strong overexpression of TRAIL mRNA in PbA-infected mice (11.36 ± 1.95 fold change; *** *p* = 0.001) in contrast to controls (1.04 ± 0.15 fold change). A strong overexpression of transcripts encoding the TRAIL protein in the renal tissue of ECM mice suggests the involvement of TRAIL in the apoptosis of renal cells, concomitantly with an acute inflammatory response elicited by IFN-γ, since this pro-inflammatory cytokine may enhance death receptor-induced apoptosis by increasing TRAIL expression locally. (**B**) Real-time RT-PCR analysis also demonstrates that CASP3 gene expression was about two-fold higher in PbA-infected mice (2.03 ± 0.18 fold change; *** *p* = 0.001) than in uninfected animals (0.99 ± 0.07 fold change). (**C**) Real-time RT-PCR analysis shows that CASP8 gene expression is significantly induced in PbA (2.13 ± 0.44; * *p* = 0.041) as compared with uninfected mice (1.03 ± 0.12 fold change). These results suggest that CASP3, which is cleaved by CASP8, may promote internucleosomal DNA fragmentation, membrane blebbing, and the formation of apoptotic bodies in the renal cells of mice with severe malaria. N = 5 for uninfected controls and PbA-infected mice. The RMCBS scores for these animals were 19.41 ± 0.25 and 11.60 ± 0.25, respectively. All animals were euthanized on day 7 post-infection, and tissue samples were harvested.

**Figure 9 pathogens-13-00376-f009:**
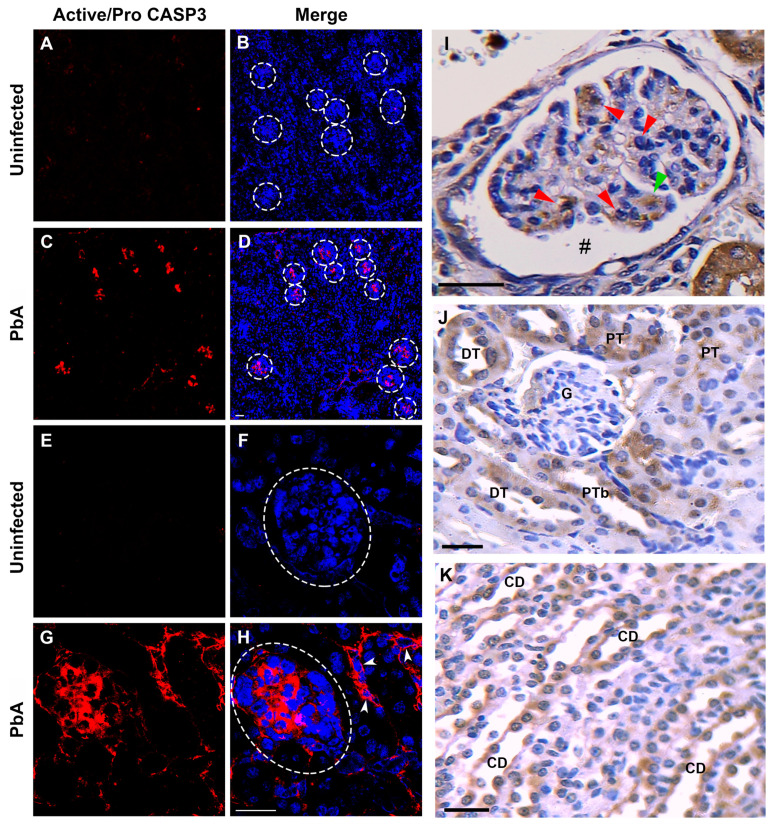
In situ immunofluorescence and immunohistochemistry assays for active/pro-Caspase-3 in kidney tissue. (**A**) Cryostat kidney section from an uninfected mouse that exhibits only a very weak auto-fluorescent signal of active/pro-CASP3 antibody (red). (**B**) Merged image with double immunolabeling for active/pro-CASP3 (red) and TOPRO-3 (blue). White-dotted circles delimit renal corpuscles. (**C**) Cryostat kidney section from a PbA-infected mouse that exhibits intense fluorescent signal of pro-CASP3 antibody in some renal corpuscles (red). (**D**) Merged image with double immunolabeling for active/pro-CASP3 (red) and TOPRO-3 (blue). (**E**) Cryostat kidney section from an uninfected mouse that shows negative signal of active/pro-CASP3 antibody. (**F**) Merged image with double immunolabeling for active/pro-CASP3 and TOPRO-3. (**G**) Cryostat kidney section from an infected mouse that exhibits strong cytoplasmic fluorescent signal of active/pro-CASP3 antibody in cells of some renal corpuscles and tubules (white head arrows). (**H**) Merged image with double immunolabeling for active/pro-CASP3 and TOPRO-3 and TOPRO-3. An exclusive and strong immunofluorescence signal of anti-active/pro-CASP3-antibody in glomerular and tubular cells of mice with ECM demonstrates that PbA infection induces apoptosis in cells from these two nephron segments. (**I**) A paraffin-embedded kidney tissue section marked with the active/pro-caspase-3 antibody revealing several positive mesangial cells (red arrow) and a positive podocyte (green arrow). An enlargement of the Bowman’s space can be observed (hashtag). (**J**,**K**) PT, DT and CD cells showing strong signal for active/pro-caspase-3. A glomerulus without collapsing and negative for active caspase-3 labeling. Tissue samples were harvested from mice euthanized at day 7 post-infection. G, glomerulus; PT, proximal convoluted tubule; DT, distal convoluted tubule; CD, collecting duct. (**A**–**H**), original magnification: ×20. (**I**–**K**), magnification: ×40. Scale bar: 40 µm.

## Data Availability

Data are contained within the article and Appendix A.

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
