# Peer review of "Involvement of Inflammatory Cytokines, Renal NaPi-IIa Cotransporter, and TRAIL Induced-Apoptosis in Experimental Malaria-Associated Acute Kidney Injury"

_pathogens, 2024, doi:10.3390/pathogens13050376_

Round 1
Reviewer 1 Report
Comments and Suggestions for Authors
Authors using the P.berghei ANKA model to .study acute kidney injury (AKI) in experimental cerebral malaria (eCM) by assessing plasma creatinine, blood urea nitrogen (BUN) and Na-dependent Pi cotransporter (NaPi-IIa) (in proximal tubules??), glomerular deposits and kidney morphology.
They also study the cytokine profile and TRAIL as a marker for apoptosis. The role of TRAIL in apoptosis in the kidney is further explored.
In the abstract, define AKI and CBA assay (maybe include a abbreviations section)
Experimental cerebral malaria is defined as eCM. In the abstract is later cerebral malaria used (written out). Be consistent in using this abbreviation. Check manuscript for consistent usage.
Material and methods are overall well described. Data are presented as Mean +/- SEM and not SD, indicative that some assessments need to be expanded, or based on another/additional measurement, e.g. RMBCS . Authors list n=5 for controls and n-6 for eCM.
Most studies list n=10 for eCM, due to attrition, as some mice will need to be euthanized because they are moribund while others don’t develop eCM. Therefore, selection of eCM mice need to be based on RMBSC and not on just day. If 100% of their mice developed eCM, this needs to be indicated, or attrition percentage.
RMBCS scale also need to be explained: what assessment were done and on what scale. RMBCS for each experiment need to me listed in the legends.
For immunocytochemistry, the dilutions of antibodies need to be included.
No material and methods included for LPS-induced endotoxemia (section 3.2)
For the magnifications, include final magnification in the pictures (use a bar) and list in the legends M= ..x). Regarding the PbA infection, the authors allow the animals to die, which starts from day 6 and runs till day 8. At that time cognitive functions declined. The mice were not euthanized when they turned moribund but allowed on, which is the usual practice. When they are moribund, they will -off course- show loss of cognitive functions. Moreover, authors report extensive hemorrhage at day 7. Additional data at an earlier time point, e.g. day 4 or 5 is needed.
Of concern is that there is no reference to approvals for animal experimentation and whether the study was approved by the Institutional Animal Care and Use Committee (IACUC). This needs to be stated, including the approval number.
The assessments are all done at day 7, when animals are moribund and no intermediate assessments done, e.g. day 4 or 5. Thus it is all these finding are no surprise. Inclusion of an intermediate timepoint is imperative.
All legends need to indicate at which day the animals were euthanized.
Figure 1: control non-infected- animals are followed till Day 24, which is not needed and does not add any additional value/information Graphs can be stopped at either day 10 or 12.
Fig 9: which cells have the caspase: A double labeling specific for the cell type and higher magnification would clarify this.
Reviewer 2 Report
Comments and Suggestions for Authors
The study investigated the mechanisms underlying acute kidney injury (AKI) in experimental cerebral malaria (ECM) induced by Plasmodium berghei. It explored the involvement of inflammatory cytokines, the renal NaPi-IIa cotransporter, and TRAIL-induced apoptosis in malaria-associated AKI. The proinflammatory response, characterized by TNF-α, IL-1β, and IL-6 release, likely contributed to kidney damage. The study also examined the role of the renal sodium/phosphate cotransporter, NaPi-IIa, which may be dysregulated during malaria infection, impacting renal dysfunction. Furthermore, TRAIL-induced apoptosis, a process where immune cells kill infected or damaged cells, was investigated as a potential contributor to AKI in malaria and was found to be upregulated. Understanding these mechanisms could provide insights into the pathogenesis of malaria-associated AKI and inform the development of therapeutic strategies for malaria-associated AKI.
The study has been performed lucidly and discussed at length.
Few references are quite outdated, and may be replaced with recent ones.
At least one reference of 2023 or 2024 may be cited.
Comments on the Quality of English LanguageThe proficiency level of the English language is satisfactory.
Round 2
Reviewer 1 Report
Comments and Suggestions for Authors
Authors responded well to the comments. Unfortunately, the authors indicated that there are no funds to purchase some antibodies and perform an addition double staining to address a localization.
alternatively, authors can show a phase contrast of the section and/or an HE staining of an adjacent section and label the specific cell types, thereby clarifying the localization. This should be done and should not be a major issue. They have the slides already, just a matter of taking an extra picture.
Author Response
Response to Reviewer 1.
We have addressed your previous recommendation in the latest revised version of the manuscript. In response to your query regarding the identification of cell types undergoing apoptosis within the renal corpuscle, we conducted an immunohistochemistry assay employing the same anti-active/pro-caspase-3 antibody utilized in the immunofluorescence assay. Our investigation was expanded to encompass other segments of the nephron across both cortical and medullary regions. The outcomes have been incorporated into Figure 9, which has been modified accordingly. A detailed methodological description of this approach has been included in section 2.8, while the results have been integrated as an extension of section 3.6. We believe that the recent suggestions have significantly enhanced the study, and we are available for any additional revisions you may need.
Round 3
Reviewer 1 Report
Comments and Suggestions for Authors
Authors responded satisfactory to the suggestions. Thank you